# Development of a Quality-of-Life Instrument to Measure Current Health Outcomes: Health-Related Quality of Life with Six Domains (HRQ-6D)

**DOI:** 10.3390/jcm12082816

**Published:** 2023-04-11

**Authors:** Mohamad Adam Bujang, Wei Hong Lai, Selvasingam Ratnasingam, Xun Ting Tiong, Yoon Khee Hon, Eileen Pin Pin Yap, Yvonne Yih Huan Jee, Nurul Fatma Diyana Ahmad, Alex Ren Jye Kim, Masliyana Husin, Jamaiyah Haniff

**Affiliations:** 1Clinical Research Centre, Sarawak General Hospital, Ministry of Health Malaysia, Kuching 93586, Sarawak, Malaysia; 2Department of Psychiatry and Mental Health, Sarawak General Hospital, Ministry of Health Malaysia, Kuching 93586, Sarawak, Malaysia; 3Institute for Clinical Research, Ministry of Health Malaysia, Block B4, National Institutes of Health (NIH), No. 1, Jalan Setia Murni U13/52, Seksyen U13, Shah Alam 40170, Selangor, Malaysia; 4Radiotherapy and Oncology Unit, Sarawak General Hospital, Ministry of Health Malaysia, Kuching 93586, Sarawak, Malaysia; 5Heart Center, Sarawak General Hospital, Ministry of Health Malaysia, Kuching-Samarahan Expressway, Kota Samarahan 94300, Sarawak, Malaysia; 6Quality Unit, Sarawak General Hospital, Ministry of Health Malaysia, Kuching 93586, Sarawak, Malaysia; 7Malaysian Health & Performance Unit, Ministry of Health Malaysia, Blok E7, Federal Government Administrative Centre, Putrajaya 62590, Selangor, Malaysia

**Keywords:** health related, quality of life, questionnaire development, reliability, validity

## Abstract

Health-related quality of life (HRQOL) is one of the most important outcome measures to be assessed by medical research. This study aims to develop and validate an instrument called the “health-related quality of life with six domains” (HRQ-6D), which aims to measure an individual’s health-related quality of life within a 24 h period of time. This is a questionnaire development study involving five phases, which are (i) to explore the subject matter content for gaining a better understanding of the topic, (ii) to develop the questionnaire, (iii) to assess both its content validity and face validity, (iv) to conduct a pilot study, and finally, (v) to undertake a field testing of the questionnaire. For the field-testing phase, a cross-sectional study involving a self-administered survey for HRQ-6D items was conducted among healthcare workers with various health conditions. Exploratory factor analysis was initially applied to construct the major dimensions of the HRQ-6D. Confirmatory factor analysis was subsequently applied to evaluate the model fit of the overall framework of the HRQ-6D. The clinical utility of this HRQ-6D was also assessed via its association with actual clinical evidence. A total of 406 respondents participated in the survey. Six domains were identified from the analysis, namely “pain”, “physical strength”, “emotion”, “self-care”, “mobility”, and “perception of future health” comprising two items in each domain. Each domain was reported to have a minimum value of Cronbach’s alpha of 0.731, and the model fit for the overall framework of the HRQ-6D was also found to be excellent. Exploratory factor analysis was undertaken for the 12 items of the HRQ-6D. All the domains can be categorized into three major dimensions, namely “health”, “body function”, and “future perception”, with a minimum value for their factor loadings of at least 0.507. A notable finding was that the HRQ-6D was significantly associated with an individual’s existing comorbidities and current status of health (*p* < 0.05). This study successfully validated the HRQ-6D, and we found it to possess both excellent levels of reliability and validity and a satisfactory degree of model fit; it was also significantly associated with actual clinical evidence.

## 1. Introduction

Health-related quality of life (HRQOL) is one of the most important outcome measures to be assessed and determined by medical research [1,2]. This parameter is an outcome based on “patient-centric” assessments that cannot be possibly measured by using a device or machine. One of the best ways to assess HRQOL is by using a specific questionnaire with adequate levels of both validity and reliability. There are many possible domains or criteria that can really contribute to the demarcation of a conceptual definition of the “HRQOL” measure for an individual, and such criteria are usually related to various factors or items of an individual’s functional and emotional well-being [3,4]. Therefore, HRQOL has now become one of the most important indicators for obtaining the outcome measure for a patient’s health condition apart from the usual routine clinical parameters such as his/her comorbidities, disease progression, and treatment complications [5,6].

One of the most popular HRQOL measures for assessing an individual’s current health condition is the EQ-5D, which is a standardized instrument for measuring the generic health status of an individual. The EQ-5D was first introduced in 1990 by the EuroQol Group [3]. The EQ-5D questionnaire is made up of two components, namely the description and evaluation of a person’s health status. The EQ-5D consists of an initial descriptive part, which provides a measurement of current health status in terms of five major domains (5D): mobility, self-care, usual activities, pain/discomfort, and anxiety/depression. Firstly, the study respondents shall self-rate their own assessment of each dimension to indicate the level of problem(s) they might possibly experience from each of the five domains by using a three-level (EQ-5D-3L) scale. Besides that, they also evaluate their overall health status by using the visual analog scale (EQ-VAS), which provides the respondent’s overall assessment of his/her health status on a scale from 0 (worst health imaginable) to 100 (best health imaginable) [4].

Various studies had proposed and recommended upgrading the EQ-5D-3L to EQ-5D-5L since this enables the new EQ-5D-5L to measure each domain along a wider continuum [7,8,9]. The similarity between these two scales is that each domain is still measured by a single rating only. For example, in the EQ-5D-5L, the study respondent should provide his/her response by selecting one item out of a total of five options. For the domain of “mobility”, these five options are arranged in a continuum of the level of problem experienced by the respondent; which can range from best to worst such as “I have no problem in walking around”, “I have slight problem in walking around”, I have moderate problem in walking around”, and “I am unable to walk around”.

Basically, the EQ-5D (EQ-5D-3L or EQ-5D-5L) was previously validated based on an assessment of its content validity by experts and also an assessment of a known-group comparison. In the known-group comparison, the EQ-5D scores from a group of study respondents with better health status were compared against the scores from another group of study respondents with poorer health status [10,11]. The results obtained from the known-group comparison demonstrated that the EQ-5D was indeed able to detect such differences in health status between the two groups of respondents. The EQ-5D was therefore identified to be a reliable and valid scale, and it was subsequently translated and validated into various languages [12,13,14,15]. Although the EQ-5D has a distinct advantage in terms of both its brevity and simplicity, it is nonetheless a difficult task to assess its validity by using statistical techniques such as exploratory factor analysis (EFA) and confirmatory factor analysis (CFA). In addition, the utility of this current HRQOL scale can be further enhanced by introducing new domains to measure the current health status of an individual person over the course of one day.

This explains why the present study is designed to develop a health-related quality of life scale that is able to measure a much broader spectrum of domains for an HRQOL measure, along with a high level of validity and reliability. Hence, the authors have named this new questionnaire “health-related quality of life with six domains” (HRQ-6D), and the primary objective of this study is to validate this by assessing both its validity and reliability.

## 2. Materials and Methods

### 2.1. Overall Process of Questionnaire Development

The entire process of developing the HRQ-6D involves a total of five phases, which are (i) to explore and understand the subject matter, (ii) to develop the questionnaire, (iii) to assess both its content validity and face validity, (iv) to conduct a pilot study, and (v) to undertake a field testing of the questionnaire [16]. Thus, the development of this scale for the measurement of the HRQ-6D is based on an observational study that utilizes both a qualitative approach (i.e., to conduct an in-depth study of the background subject matter for the purpose of item generation, develop a scale that includes all the items, and finally assess both its content and face validity) and a quantitative approach (i.e., to conduct an initial pilot study followed by a field test).

### 2.2. Review of the Background Subject Matter

The study began by taking the first step to conduct a literature review of the published studies that are related to HRQOL. This was carried out to enhance the level of our understanding of the subject matter through an exploration of the underpinning theories and the related concepts of HRQOL before developing the overall conceptual framework for the HRQ-6D. Upon the identification of a list of core components that encompass the overall concept of HRQOL, the authors then proceeded to develop the items that were further categorized into six different domains. The domains of the HRQ-6D consisted of “pain”, “physical energy”, “mobility”, “self-care”, “emotion”, and “perception for future health” [4,17,18,19,20,21,22,23]. All these domains are almost similar to those of the EQ-5D. New domains such as “physical energy” and “perception for future health” were incorporated into the HRQ-6D.

### 2.3. Development of an Overall Framework for Item Generation for HRQ-6D

The overall aim of the HRQ-6D is to measure how the study respondents feel about their health condition today, which is expressed in terms of a wide variety of quality-of-life indicators. Thus, the respondents were asked about the extent to which certain events might have actually or possibly occurred as part of their health condition today. An interval scale that was based on a five-point Likert scale, i.e., “strongly disagree”, “disagree”, “neutral”, “agree” and “strongly agree”, was used in this study. Previously, we predetermined the HRQ-6D comprising six domains, and two items were designed for each domain. Thus, altogether, there were 12 items in the HRQ-6D.

### 2.4. Content Validity and Face Validity

The content validity of the HRQ-6D was assessed by five subject matter experts (SMEs), and this panel of SMEs included an epidemiologist, a senior medical officer, a psychiatrist, and two senior researchers with Ph.D. qualifications. The overall content validity was assessed using two approaches: qualitative and quantitative methods. For a qualitative approach, all the comments were given due consideration by the researcher, who made further amendments to the questionnaire by improving the appropriateness of the syntax or wording of these questions in order to maintain content validity and also the overall quality and accuracy of the content of such questions or items. This was an iterative process, and it was repeated several times until the overall results were found to be satisfactory.

For a quantitative approach, the elected panels rated each item in terms of its content validity, and then the content validity index (CVI) was calculated. All the items were considered to be valid after the CVI reached a minimum of 0.99. All the HRQ-6D items were initially developed in English. As English is a second language in Malaysia, it was necessary for a language expert to determine that the syntax and wording of all the items on the HRQ-6D were appropriately used to ensure that all these items were expressed by the correct usage of English grammar and vocabulary. Subsequently, all the questions were translated into the Malay language by using both forward and backward translations. The study group carefully reviewed both the English and Malay versions of the questionnaire, and after making all necessary revisions for both versions of the questionnaire, an assessment of its face validity was conducted.

The final draft of the questionnaire was then pretested by a group of ten healthcare workers in order to determine its face validity. Each of these healthcare workers was personally informed about the proposed scope of the study, the adequacy of the provision of specific instructions for filling in their responses on the instrument, the response format itself, and the meaning of all the items on the instrument. There were basically two important measures to assess the face validity of this questionnaire that had to be vetted at this stage: the first is the measurable extent of each item for defining a particular trait, and the second is a carefully assembled set of items that can collectively define a trait and serve as a representative of all the aspects of the trait.

A cognitive debriefing process was conducted for the purpose of linguistic validation. The participants were then asked several questions, for instance, what they thought the item was asking, whether they were able to reformulate the item in their own words and explain how they arrived at their answers, which particular words they might have associated with an item, whether or not the item demonstrated clarity and coherence in meaning, and if there were any words they did not understand or found to have been inappropriately used. If alternative choices for such words were available, the study participants would then be given an opportunity to offer various suggestions for such alternative words so that they could understand the true meaning of these items or questions that had originally been intended by the researchers. All these findings will be compiled together for the preparation of a final report, which will include all the relevant details about the study participants, as well as a list of final recommendations that were proposed for improving the overall content of a study instrument.

### 2.5. Pilot Study

After all the results based on the assessment of both content validity and face validity were determined to be satisfactory, a pilot study was conducted among 30 respondents who were healthcare workers from various sociodemographic backgrounds and exhibiting a wide variety of different health statuses due to a broad range of current medical conditions and past medical history records. The aim of the pilot study was to test the reliability of HRQ-6D items and their domains. Its test–retest reliability was determined within a lag time of two weeks. Thus, the overall time period for conducting a pilot study consists of three weeks, which includes both the recruitment period and the lag time. The target was to achieve a minimum value of 0.90 for the kappa coefficient of each item for determining the strength of agreement between test and retest and also a minimum value of 0.65 for Cronbach’s alpha to check the internal consistency or reliability of the targeted domains. After the results of all these reliability tests were found to be satisfactory, a field test was conducted to validate the HRQ-6D.

### 2.6. Field Testing and Study Design

The field testing of the HRQ-6D was conducted via a cross-sectional study that included a self-administered survey. The study sample was recruited from Sarawak General Hospital and The Heart Centre, Sarawak General Hospital. Both are governmental healthcare facilities under the administration of the Ministry of Health, Malaysia. The selection criteria included (i) all workers who are currently working in a healthcare setting, including permanent, contract, and temporary staff, irrespective of whether they are from the government or the private sectors; (ii) all those 18 years old and above; and (iii) those who agreed to participate in the study. These respondents were exhibiting various conditions of current health status. Nevertheless, those study respondents who were unconscious, too sick, in a comatose state, of unsound mind, or with an unstable mental condition during the recruitment period were excluded from this study.

### 2.7. Instrumentation for a Self-Assessment of Health-Related Quality of Life

Apart from the usual questions on the demographic profile and the HRQ-6D with 12 items, we also asked questions related to health conditions. The first questions pertaining to an assessment of current health conditions aimed to elicit information regarding respondents’ status of comorbidities, such as diabetes mellitus, dyslipidemia, hypertension, and others. The second question was related to an evaluation of health status. For this question, five different categories were created, and respondents needed to choose only one category that best described his/her health status. The five categories are described as follows:Category 1

I am healthy (never been diagnosed with any medical condition except mild fever or headache, and I have never been hospitalized except for child delivery);
Category 2

I have been diagnosed with one or more than one disease but never been hospitalized (except for child delivery);
Category 3

I have been diagnosed with one or more than one disease and have been hospitalized due to disease progression or complications;
Category 4

I have been diagnosed with one or more than one disease and have been hospitalized more than 3 times due to disease progression and complications;
Category 5

I am dependent on medicine and/or medical procedure(s) and/or medical equipment to keep me alive (e.g., major surgery, renal dialysis, blood transfusion for thalassemia, heart transplant/stent placement, chemotherapy cycles for cancer treatment, etc.).

In addition, there was one question regarding a self-rated assessment for health-related quality of life for today (within 24 h). The respondents were required to describe their HRQOL for today based on a specific scale by circling any number from 0 (poorest health HRQOL) to 10 (excellent HRQOL).

### 2.8. Data Collection

All data were collected from the study respondents for two months from February 2022 until April 2022. The recruitment of respondents was based on a voluntary basis. A snowball sampling technique was adopted to retrieve a sample of respondents from the study population, which involved contacting the prospective respondents via email or WhatsApp. The researchers initially sent an email or a WhatsApp message to each department, and the head (or manager) of each department would then disseminate such information to their staff and also invite them to respond to this questionnaire by providing a link to the google form of the HRQ-6D (in order for them to fill in their responses on this questionnaire).

### 2.9. Sample Size Planning

The determination of the minimum required sample size was based on a rule-of-thumb for exploratory factor analysis (EFA). There are 12 items for the HRQ-6D. This means that based on the rule of thumb of using the 10:1 ratio for sample size determination, the minimum sample size of 12 × 10 = 120 respondents would be required [24,25]. To make allowances for a non-response rate of 20.0%, the minimum sample size required was inflated to at least 150 respondents.

### 2.10. Statistical Analysis

The ultimate aim of this study was to develop a new questionnaire that can accurately assess the current HRQOL of a person. Descriptive statistics were used to describe the sociodemographic profiles and the current status of health conditions of all the study respondents who had participated in this study. Kappa agreement and Cronbach’s alpha were applied to measure the reliability of the HRQ-6D. Confirmatory factor analysis (CFA) was applied to evaluate the model fit of the HRQ-6D based on indicators such as the chi-square test (<3.0), comparative fit index (CFI > 0.90), root-mean-square error approximation (RMSEA < 0.08) and standardized root-mean-square residual (SRMR < 0.08) were estimated [26].

Exploratory factor analysis (EFA) was conducted to determine the total number of dimensions (or major categories) of the HRQ-6D. In our analysis, we utilized principal axis factoring (PAF) as the method of factor extraction together with the Promax rotation method. A factor solution was derived by adhering to Kaiser’s criterion, which stipulated that only those factors with an eigenvalue of >1 would be retained. Meanwhile, an independent sample *t*-test, one-way analysis of variance test, and Pearson’s correlation test were also applied to determine the level of association or correlation between the newly developed HRQ-6D and an individual’s current health status. All the analyses were conducted by using SPSS (IBM Corp. Released 2011. IBM SPSS Statistics for Windows, Version 20.0. Armonk, NY, USA) and R Core Team (2013) (R: A language and environment for statistical computing. R Foundation for Statistical Computing, Vienna, Austria. URL http://www.R-project.org/ (accessed on 15 October 2022)).

### 2.11. Ethical and Regulatory Considerations

Prior written informed consent was obtained from all the respondents by providing them with an informed consent form along with the online version of the questionnaire. Only those study respondents who had given informed consent for participating in this study were surveyed. This study adhered to all the relevant guidelines and regulations that are stipulated by the Medical Research and Ethics Committee (MREC), National Institutes of Health, Ministry of Health Malaysia. Ethical approval for this study was granted by Medical Research and Ethics Committee (MREC), and the permission for publication of study findings was granted by the Director General, Ministry of Health, Malaysia.

## 3. Results

### 3.1. Sample Population

A total of 406 respondents participated in the survey. The majority of these respondents were female (81.8), aged between 18 and 35 years (53.0%), had graduated with either a certificate or a diploma (65.3%), were married (68.0%), and belonged to the nursing profession (63.3%). Notably, 29.3% of the respondents had at least one comorbidity, and a majority of them had hypertension (8.9%), dyslipidemia (8.1%), and diabetes mellitus (3.4%). In terms of the category of health status, a majority of them were considered “healthy” (Category 1) (70.6%), a small proportion of them with at least one comorbidity (Category 2) (23.9%), and a minority of them had been hospitalized due to disease progression and other associated complications (Category 3, 4, and 5) (5.4%) (Table 1).

### 3.2. Reliability and Validity of HRQ-6D

The value of Cronbach’s alpha for the HRQ-6D domains was calculated to range between 0.821 and 0.909. Cronbach’s alpha for all the items in the HRQ-6D was calculated to be 0.904. Further analysis was conducted to evaluate the model fit of the HRQ-6D. The degree of model fit is an important consideration because it enables us to assess whether each domain demonstrates its own significant or “special” meaning and also whether it represents a separate entity that can be measured independently from all the other domain(s). The summary of the results obtained by conducting CFA revealed that the HRQ-6D had an excellent model fit based on the calculated values of the four fitness indicators, i.e., chi-square test < 0.30, CFI > 0.90, RMSEA < 0.80, and SRMR < 0.08 (Table 2).

Based on the results obtained from the EFA, there were a total of three dimensions that could be constructed from the analysis. Dimension 1 with six items consisted of all the items from the three domains of “pain”, “physical energy”, and “emotion”. Dimension 2 with four items consisted of two items from the two domains “mobility” and “self-care”, respectively, while Dimension 3 consisted of two items from the domain “perception of future health”. The minimum value of Cronbach’s alpha for all the dimensions was found to be more than 0.888. The summary of the results obtained from the EFA revealed the minimum factor loading to be 0.507 (Table 3). Upon a careful examination of the actual content for each of these items, our study team finally decided to define Domain 1 as “health”, Domain 2 as “body function” and Domain 3 as “perception”. The proposed scoring mechanism is presented in Table 4, and the final scale for the HRQ-6D and the self-rated HRQOL are presented in Appendix A.

### 3.3. Clinical Relevance of HRQ-6D

We found that the status of comorbidity was significantly associated with the standard score of the HRQ-6D (*p* = 0.013), the health dimension (*p* = 0.011), and the perception dimension (*p* = 0.010). The current health status was associated with all the dimensions and the standard score of the HRQ-6D (health dimension, *p* < 0.001; body function dimension, *p* = 0.019; perception dimension, *p* < 0.001; the standard score of the HRQ-6D, *p* < 0.001) (Table 5). In addition, the self-rated HRQOL from 0 (poorest HRQOL) to 10 (excellent HRQOL) was also found to correlate very well with the HRQ-6D and with all its dimensions (*p* < 0.001) (Table 6).

## 4. Discussion

### 4.1. Domains and Dimensions of HRQ-6D

This study found that the HRQ-6D is a reliable and valid scale to measure the current health status of an individual. The HRQ-6D is measured based on six domains, consisting of “pain” “physical strength”, “emotion”, “mobility”, “self-care”, and “perception of future health”. This scale was almost identical to that of the EQ-5D, with the exception that it consisted of a few more items for describing and delineating the target domain, and it also included a few additional domains, namely ‘physical energy’ and ‘perception of future health’. Since each domain consisted of more than one item, it was possible to conduct the validity test based on the EFA and CFA, and it was subsequently found that the results obtained for determining the construct validity with both EFA and CFA were excellent. This indicates that in contrast to the EQ-5D, the “HRQ-6D” accrued additional evidence in terms of its statistical validity.

Unlike the EQ-5D, the domain of “usual activities” was not considered for incorporation into the HRQ-6D since this domain is not specific and in fact can measure various unrelated activities such as study, work, leisure, etc. [4]. Some respondents might find it too difficult to respond to these questions because these responses are often not objectively determined. For example, studying may require a minimal level of activity for a typical student; however, certain jobs can demand a much higher level of intensity for an activity for a typical working adult. In addition, the domain “usual activities” can also refer to unhealthy activities such as playing online games or drinking alcohol with peers. Since different people might have engaged in so many different types of usual activities, the respondents might be confused about which types of usual activities should be measured in this case. Therefore, the study team finally decided not to include the domain “usual activities” in the HRQ-6D because it did not demonstrate its specificity for representing the overall health measure.

The HRQ-6D consists of a total of six domains that are relevant to accurately measure the current status of HRQOL. This scale can therefore be used for routine clinical practice, for instance, during an assessment of patients’ current health outcome(s), and also in research work, for instance, to measure the clinical effectiveness of an intervention. The overall score for the HRQ-6D can therefore be aptly utilized to represent the overall health HRQOL and the three dimensions collectively represent the HRQOL in terms of an individual’s health, body function, and perception. These three dimensions are divided into six individual domains, which collectively provide a more specific measurement of HRQOL, thus serving as a valid tool within clinical and research contexts. All these dimensions are supported by the existing literature, and in this study, they were derived from the full extent of the overall “quality-of-life” concept introduced by the EQ-5D [4,17,18,19,20,21]. This newly developed instrument thereby further augments the clinical utility of the EQ-5D by incorporating various additional measures such as physical energy and perception of future health [17,22,23].

### 4.2. Correlation with Clinical Evidence

A scale of HRQOL such as the EQ-5D should demonstrate adequate sensitivity or responsiveness to the differences between healthy and non-healthy people so that it can be used to distinguish between the two [5,6]. Besides statistical evidence in terms of internal consistency, construct, and model fit, in this study, we also found that the HRQ-6D and its dimensions are sensitive enough to differentiate between healthy and non-healthy people. In this study, all the respondents were healthcare workers who were still actively working, but some of them already experienced some degree of “defect” in their health conditions. Health defect is measured based on the presence of comorbidity and current health status from a collective assessment of five different categories. The results show that patients who present with comorbidities often experience a poorer health-related quality of life.

Furthermore, the HRQ-6D has an additional utility by virtue of its ability to differentiate those who are healthy (Category 1), those who have been inflicted by some diseases but have never been hospitalized due to disease progression or other associated complications (Category 2), and those who have already experienced some disease progression or other associated complications (Categories 3, 4, and 5). Collectively, there were very few respondents who belonged to Categories 4 and 5, so they were both included in Category 3. In addition, the respondents’ self-rated HRQOL (from a scale of 0 to 10) was also found to correlate very well with the scale of the HRQ-6D along with its three dimensions. This finding is also consistent with that of the EQ-5D, wherein the corresponding score for the EQ-5D also demonstrated a high level of correlation with its VAS score [6]. All these findings indicate that cumulative clinical evidence points to the clinical relevance of the HRQ-6D. This study, therefore, demonstrates that clinical evidence does correlate with the HRQ-6D, which deems it to be suitable for measuring the HRQOL of both healthy individuals and real patients.

### 4.3. Future Studies

As the HRQ-6D has just been developed, its scale has to be applied in future research to determine to what extent HRQ-6D is able to fit the criteria necessary for obtaining an accurate measurement of HRQOL. First and foremost, in order to further improve the effectiveness of the HRQOL measure, the HRQ-6D should possess a few desirable properties. Such properties may include the validity of the measure; appropriateness; and acceptability especially for clinical practice; reliability or stability; ability to be responsive to change; and ease of interpretability [27]. Therefore, this study recommends future researchers enhance the wide applicability of the HRQ-6D by translating it into various other languages and also applying it in real-life clinical practice and research.

### 4.4. Scoring Mechanism

In the HRQ-6D, all items are assumed to have similar weights, and therefore, a simple scoring mechanism can be applied to estimate an overall score for the HRQ-6D, which comprises the three major dimensions and the six domains. Each item is measured using a five-point Likert scale as follows: strongly disagree = 5, disagree = 4, neutral = 3, agree = 2, and strongly agree = 1. Therefore, a higher score will reflect a better HRQOL. Then, a summation can be performed for calculating an overall score for the HRQ-6D via an assessment of all three dimensions and six domains. All the scores can then be converted into a percentage to standardize all the scores. This facilitates the ease of interpretability of the HRQ-6D by ensuring that a higher score would indicate a better HRQOL of a person. The simple calculation of the scoring mechanism is presented in Table 4.

### 4.5. Limitations of Study

This study only included healthcare workers as its study sample. The ideal study sample should be the general population with various health conditions. However, the general population is too large, and it would also be very difficult to sample due to the dispersed or scattered nature of population distribution. Hence, it would be too difficult to conduct the survey based on a set target, especially during the COVID-19 pandemic. Therefore, the study sample was obtained from healthcare workers with the initial assumption that the characteristics of the sample consisted of various health conditions and would be almost similar to the ideal target-study population, which is in fact the general population. The favorable results from this study thus justified that the items and domains of the HRQ-6D be developed in line with those of the existing literature.

### 4.6. Conclusions

The health-related quality of life with six domains (HRQ-6D) was successfully validated with an excellent level of reliability, validity, and model fit. The HRQ-6D consists of 3 dimensions, 6 domains, and 12 items. It can be used to measure an individual’s current HRQOL over the course of one day. All the items, domains, and three dimensions of the HRQ-6D were validated based on their content validity, face validity, and the statistical evidence accrued. Moreover, the total score of the HRQ-6D and its dimensions (health, body function, and perception) were also found to be associated with clinical evidence, thereby demonstrating its clinical utility.

## Figures and Tables

**Table 1 jcm-12-02816-t001:** Demographic and clinical profile of respondents.

Profile		n	%
Gender	Male	74	18.2
	Female	332	81.8
Age group	18 to 35 years	215	53.0
	36 to 40 years	88	21.7
	41 to 50 years	89	21.9
	51 to 60 years	14	3.5
Marital status	Married	276	68.0
	Single and never married	114	28.1
	Spouse passed away/Divorcee	16	3.9
Occupation	Specialist	21	5.2
	Medical doctor	50	12.3
	Pharmacist	16	3.9
	Nurses	257	63.3
	Assistant medical doctor	11	2.7
	Other allied health officer	33	8.1
	Others	18	4.5
Status of comorbidity	With comorbidities	119	29.3
	Without any comorbidities	287	70.7
Specific comorbidity ^a^	Diabetes mellitus	14	3.4
	Dyslipidemia	33	8.1
	Hypertension	36	8.9
	Chronic heart disease	3	0.7
	Stroke	1	0.2
	Chronic kidney disease	3	0.7
	Thyroid	11	2.7
	Asthma	18	4.4
	Gout	5	1.2
	Cancer	6	1.5
	Others	42	10.3
Health’s status	Category 1	284	70.6
	Category 2	96	23.9
	Category 3	17	4.2
	Category 4	2	0.5
	Category 5	3	0.7

^a^ Each respondent can have more than one comorbidity.

**Table 2 jcm-12-02816-t002:** Summary of results based on Cronbach’s alpha and confirmatory factor analysis for HRQ-6D.

	Cronbach’s Alpha	Chi-Square Test (<3.0)	CFI > 0.90	RMSEA < 0.08	SRMR < 0.08
Model fit of HRQ-6D		0.246	0.998	0.056	0.066
Pain	0.821				
Physical energy	0.886				
Emotion	0.871				
Mobility	0.869				
Self-care	0.851				
Perception of future health	0.909				
All items	0.904				

**Table 3 jcm-12-02816-t003:** Summary of results based on Cronbach’s alpha and exploratory factor analysis for HRQ-6D.

Items	(Dimensions)			Cronbach’s Alpha
	Health	Body Function	Perception	
I feel lack of physical energy	0.900			0.888
I feel tired even at rest	0.881			
I feel unhealthy	0.866			
I feel pain at any part of my body	0.643			
I feel depressed	0.536			
I feel anxious	0.507			
I have difficulty to move from one place to another		0.984		0.918
I have problem attending to my self-care needs		0.899		
I have problem doing household chores		0.781		
My movements are slower than people of my age		0.768		
I am worried that I will suffer poor health within 5 years			0.948	0.904
I am worried that my lifespan is shorter than people of my age			0.813	
Extraction method: principal axis factoring.				
Rotation method: Promax with Kaiser normalization.				

**Table 4 jcm-12-02816-t004:** A scoring mechanism for HRQ-6D.

Dimensions	Domains	Item	Scoring Mechanism
Health	Pain	Q1,Q2	[(Q1 + Q2)/10] × 100
	Physical energy	Q3,Q4	[(Q3 + Q4)/10] × 100
	Emotion	Q5,Q6	[(Q5 + Q6)/10] × 100
		Q1 until Q6	[(Q1 + Q2 + … + Q6)/30] × 100
Body function	Mobility	Q7,Q8	[(Q7 + Q8)/10] × 100
	Self-care	Q9,Q10	[(Q9 + Q10)/10] × 100
		Q7 until Q10	[(Q7 + Q8 + … + Q10)/20] × 100
Perception	Perception of future health	Q11,Q12	[(Q11 + Q12)/10] × 100
Total		Q1 until Q12	[(Q1 + Q2 + … + Q12)/60] × 100

**Table 5 jcm-12-02816-t005:** Association of HRQ-6D with the status of comorbidity and health conditions.

Dimension/Total	Status of Health	n	Mean	SD	*p*-Value
Status of comorbidity					
Health	No	290	77.5	17.2	0.011
	Yes	116	72.6	18.0	
Body function	No	290	91.6	13.1	0.489
	Yes	116	90.5	14.8	
Perception	No	290	73.4	23.8	0.010
	Yes	116	66.5	25.1	
HRQ-6D	No	290	81.5	14.0	0.013
	Yes	116	77.6	14.9	
Based on the self-assessment of health condition					
Health	Category 1	284	77.7	17.1	<0.001
	Category 2	96	74.3	18.2	
	Category 3–5	22	62.7	13.2	
Body Function	Category 1	284	91.9	12.7	0.019
	Category 2	96	91.0	14.8	
	Category 3–5	22	83.4	17.7	
Perception	Category 1	284	73.8	23.6	<0.001
	Category 2	96	69.9	24.0	
	Category 3–5	22	45.9	21.5	
HRQ-6D	Category 1	284	81.7	13.8	<0.001
	Category 2	96	79.2	15.1	
	Category 3–5	22	66.8	11.2	

Note: Category 1: healthy and without comorbidity; Category 2: with comorbidity(ies) but never been hospitalized due to disease progression or complications; Category 3: with comorbidity(ies) but already been hospitalized due to disease progression or complications; Category 4: with comorbidity(ies) but already been hospitalized for at least three times due to disease progression or complications; Category 5: dependent on medicine and/or medical procedure(s) and/or medical equipment to keep alive (i.e., major surgery, dialysis, blood transfusion for thalassemia, heart transplant/stenting, chemotherapy for cancer, etc.).

**Table 6 jcm-12-02816-t006:** Correlation between self-rated HRQOL and HRQ-6D.

		Health	Body Function	Perception	HRQ-6D
Self-rated HRQOL (1 to 10)	Coefficient	0.628	0.317	0.389	0.593
	*p*-value	*p* < 0.001	*p* < 0.001	*p* < 0.001	*p* < 0.001

## Data Availability

The data presented in this study are available on request from the corresponding author.

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
