# Peer review of "Development of a Quality-of-Life Instrument to Measure Current Health Outcomes: Health-Related Quality of Life with Six Domains (HRQ-6D)"

_jcm, 2023, doi:10.3390/jcm12082816_

Round 1
Reviewer 1 Report
The manuscript, Health-related Quality of Life with Six Dimensions (HRQ-6D), expands a currently used and validated instrument which measures the general health and quality of an individual within a 24-hour population. Unlike prior versions of the survey the new one expands to include dimensions on physical energy and perception for the future health and validates within a population of healthcare workers.
The current article would benefit from some discussion of the objective of the study to assess within a convenient population of health workers. The introduction does not prepare a reader for how the original HQROL 5-D and 5-L were validated. The problem with interpretation of the data e.g. whether they are validated are the fact these were health care workers who may be more aware of their health and co-morbidities used to validate the survey, have a better sense of perceived health and more willing to participate in a survey. Moreover the population is fairly young, female and married.
Within the discussion, the authors may want to briefly mention whether the unexpanded HRQOL have been validated for populations outside Europe. Are there any cultural differences between cultures in Malaysia and in the Europe that would alter its performance?
1. Introduction: The validation never explicitly states that the survey is intended for use within a population of health care workers. The current introduction could be re-written, shortened to accommodate why the authors want to expand the HQROL, how previous surveys have been validated and whether these instruments have previously been translated and used in non-European nations?
2. Materials and Methods, Section 2.6: May be good to mention the time period the pilot was launched to understand the exclusions.
3. Materials and Methods Section 4.2, page 9: Purpose is to assess healthy vs non-healthy people; can you really have a validated instrument using a working population of healthcare workers who are already healthy?
4. Discussion: The first sentence starts with a reliable and valid scale to measure the current health of individuals – based on a population of health care workers. The statistical validity and use of EFA/CFA are not the issue; rather it is the validity within a general population. If the authors would like to be more specific of how their validation worked e.g. within a population of health care workers who are likely healthier than general populations.
5. Discussion, Section 4.1, paragraph 2: Usual activities might an important feature to address general health --- Some problems with performing their usual activities might be an indicator of poor mental or physical health.
Author Response
Comments and Suggestions for Authors (Reviewer 1)
Comment 1
The manuscript, Health-related Quality of Life with Six Dimensions (HRQ-6D), expands a currently used and validated instrument which measures the general health and quality of an individual within a 24-hour population. Unlike prior versions of the survey the new one expands to include dimensions on physical energy and perception for the future health and validates within a population of healthcare workers.
The current article would benefit from some discussion of the objective of the study to assess within a convenient population of health workers. The introduction does not prepare a reader for how the original HQROL 5-D and 5-L were validated. The problem with the interpretation of the data e.g. whether they are validated is the fact these were healthcare workers who may be more aware of their health and co-morbidities used to validate the survey, have a better sense of perceived health, and are more willing to participate in a survey. Moreover, the population is fairly young, female, and married.
Reply
The EQ-5D/5L was validated based on content expert and clinical evidence. The authors have included a statement and cited two additional sources of reference for emphasizing this point in the main text of this manuscript. Kindly refer to paragraph no. 4 in the introduction section and two additional references 10 and 11 in the reference list (highlighted in yellow).
HRQ-6D follows a similar approach as that EQ-5D. Furthermore, apart from an assessment of content validity, HRQ-6D has also been validated based on clinical evidence. The clinical evidence of HRQ-6D was accrued by basing on each respondent’s comorbidities and self-report health status. The responses elicited from the healthcare workers regarding their own health are reliable since they are fully aware of their own health conditions and the presence of any other co-morbidities (if any). Furthermore, statistical evidence has also been ascribed to HRQ-6D as discussed in this paper.
In a future study, we intend to expand the scope of utilization of HRQ-6D by extrapolating its use to other research studies in which the same study group shall comprise real patients who are presenting with different medical conditions and clinical diagnoses. In this study, we will then be able to demonstrate that HRQ-6D is also useful for evaluating the health status among real patients who are presenting with various disease conditions. Thank you.
Comment 2
Within the discussion, the authors may want to briefly mention whether the unexpanded HRQOL have been validated for populations outside Europe. Are there any cultural differences between cultures in Malaysia and in Europe that would alter its performance?
Reply
EQ-5D is a validated scale that has also been translated and validated in various languages. The authors have cited four additional references to support the claim above. Therefore, any possible variation in the utility of EQ-5D which may have arisen from cultural differences between study populations from different parts of the world would have been fully accounted for, since the rigorous process of validating a translated version of EQ-5D would be able to sufficiently cater for the specific needs of diverse study populations from different parts of the world (with differing language and socio-cultural backgrounds). Kindly refer to paragraph no. 4 in the introduction section and references 12, 13, 14, and 15 in the reference list (highlighted in yellow). Thank you.
Comment 3
Introduction: The validation never explicitly states that the survey is intended for use within a population of health care workers. The current introduction could be re-written, shortened to accommodate why the authors want to expand the HQROL, how previous surveys have been validated and whether these instruments have previously been translated and used in non-European nations?
Reply
The authors intend to use the HRQ-6D for measuring health status among healthy and non-healthy participants. At this early stage, the authors aim to validate HRQ-6D among a group of healthcare workers who comprise a mixture of both healthy and non-healthy respondents. In a future study, we intend to enhance the utility of HRQ-6D by conducting other research studies in which real patients who present with a range of diverse medical conditions and clinical diagnose are being recruited as the study participants. By doing so, we will be able to demonstrate that HRQ-6D is also useful for evaluating health status among a study population with different disease conditions.
The main justification for the authors to develop this new ‘HRQ-6D’ is that we intend to introduce several new domains which are not included in EQ-5D but are equally important in measuring health status. We have clearly stated the above justification in both the introduction and discussion sections of this paper.
On a similar note, the EQ-5D/5L was validated based on content expert and clinical evidence. The authors have cited two additional sources of reference in support of this. In tandem with EQ-5D, the HRQ-6D also follows a similar approach as that of EQ-5D by basing its validity on an assessment of both its content validity and clinical evidence. The clinical evidence was accrued by basing on each respondent’s comorbidities and self-report health status. In addition, our study has conferred additional statistical evidence for the utility of HRQ-6D as discussed in both the results and discussion sections of this paper.
As EQ-5D is validated that has also been translated and validated in various languages; hence, it has been designed to be able to cater and to accommodate the specific needs of a diverse group of study populations who exhibit differing language and socio-cultural backgrounds since they are from many different parts of the world. Kindly refer to paragraph 4 in the introduction section of this paper. Thank you.
Comment 4
Materials and Methods, Section 2.6: May be good to mention the time period the pilot was launched to understand the exclusions.
Reply
Noted. The authors have included an additional statement to address this concern, which reads: “Thus, the overall time period for a pilot study is three weeks which includes the recruitment period and the lag time.” Please refer to the paragraph in section 2.5 (highlighted in yellow). Thank you.
Comment 5
Materials and Methods Section 4.2, page 9: Purpose is to assess healthy vs non-healthy people; can you really have a validated instrument using a working population of healthcare workers who are already healthy?
Reply
For the purpose of this study, the authors have specifically developed a set of new questions to identify a list of possible comorbidities and the current health status of an individual person from among all the study participants. These new questions are also self-report items, which are designed to elicit descriptions of the self-reported comorbidities and current health status. Thank you.
Comment 6
Discussion: The first sentence starts with a reliable and valid scale to measure the current health of individuals – based on a population of health care workers. The statistical validity and use of EFA/CFA are not the issue; rather it is the validity within a general population. If the authors would like to be more specific of how their validation worked e.g. within a population of health care workers who are likely healthier than general populations.
Reply
The authors of this paper originally intended to develop the HRQ-6D ideally for use to measure health status among both healthy and non-healthy participants. Hence, for the purpose of this paper, we have presumed that a population of healthcare workers has an almost identical health status as that of a general population. Therefore, the authors have clearly stated that such a presumption is one of the limitations of this study.
In a future study, we intend to extrapolate the use of the new HRQ-6D to real patients who may present a list of different current diagnoses, by expanding the scale of this study to also include the recruitment of real patients seeking medical care who will present with differing current medical conditions and clinical diagnoses. Henceforth, we shall be able to demonstrate the utility of HRQ-6D for evaluating the health status among a study population who might present with different types of diseases and/or medical conditions. Thank you.
Comment 7
Discussion, Section 4.1, paragraph 2: Usual activities might an important feature to address general health --- Some problems with performing their usual activities might be an indicator of poor mental or physical health.
Reply
Noted. One of the main reasons for the authors to have decided to drop the item ‘usual activities’ from HRQ-6D is that the definition of usual activities can possibly include a very broad scope of activities which often vary in terms of their nature, type, and intensity; most of which can also be totally irrelevant to us. The authors have provided a detailed explanation for this decision in the discussion section of this paper. Moreover, the HRQ-6D has been designed to include two specific domains such as physical strength and mobility which can also be referred to as the ‘usual activities’ performed by an individual (since many of these usual activities will involve the expenditure of both strength and mobility). Thank you.
Reviewer 2 Report
Manuscript Id : jcm-2223948
Title: Development of Quality of Life Instrument to Measure Current Health Outcome, Health-related Quality of Life with Six Domains (HRQ-6D)
This study designed, developed and tested an improved an instrument called.
‘Health-related Quality of Life with Six Domains’ (HRQ-6D) to measures the Health-related Quality of Life (HRQOL). The authors claimed that the proposed instrument is better than a similar existing measure, EQ5D, for the reason that the new measure uses 6 domains where as EQ5D uses only 5 domains. The manuscript is well written as presented, except for a few unclear points.
The vigorous effort that the research team put on to design, test the questionnaire as well as collect the data amidst of the COVID-19 pandemic is admirable. Nevertheless, the data are gathered only from the health staff, and it will be weakness of this study. Moreover, there are some mis-matching points in the questionnaire. For instance, if a patient visiting a clinic will select the answer as ‘Strongly disagree’ to Q1, if he/she does not experience ay pain in the body. However, the same patient may feel very tired and hence will feel unhealthy. Therefore, this patient will select ‘Strongly agree’ to Q2.
Another instance is Q7 & Q8. Some people are lazy by nature. Movements of such a person may be slower than the same-aged people. However. This lazy person does not have any difficulty to move from one place to another.
A complete statistical analysis was carried out in this study and the results are well interpreted.
Nonetheless, there is one unclear sentence. In the 2nd paragraph of the Sub-section 2.10, it is mentioned that ‘an Independent sample t-test, One-way Analysis of Variance test’ was applied. This statement needs further explanation and connects to the results.
Author Response
Comments and Suggestions for Authors (Reviewer 2)
Comment 1
This study designed, developed and tested an improved an instrument called. ‘Health-related Quality of Life with Six Domains’ (HRQ-6D) to measures the Health-related Quality of Life (HRQOL). The authors claimed that the proposed instrument is better than a similar existing measure, EQ5D, for the reason that the new measure uses 6 domains where as EQ5D uses only 5 domains. The manuscript is well written as presented, except for a few unclear points.
The vigorous effort that the research team put on to design, test the questionnaire as well as collect the data amidst of the COVID-19 pandemic is admirable. Nevertheless, the data are gathered only from the health staff, and it will be weakness of this study. Moreover, there are some mis-matching points in the questionnaire. For instance, if a patient visiting a clinic will select the answer as ‘Strongly disagree’ to Q1, if he/she does not experience any pain in the body. However, the same patient may feel very tired and hence will feel unhealthy. Therefore, this patient will select ‘Strongly agree’ to Q2.
Reply
Thank you for the compliment. Ideally, the authors have intended to use HRQ-6D for measuring the current health status among both healthy and non-healthy participants. In this study, we have presumed a priori that a population of health care workers to have exhibited a similar spectrum of medical conditions to that of a general population. Hence, the authors have given due consideration of this a priori presumption by stating it as one of the major limitations of this study. At this early stage, the authors aim to validate HRQ-6D among a group of health care workers who comprise a mixture of both healthy and non-healthy respondents.
In a future study, we intend to enhance the utility of HRQ-6D by conducting other research studies in which real patients presenting with a range of diverse medical conditions and clinical diagnose are being recruited as the study participants. By doing so, we will be able to demonstrate the utility of HRQ-6D for evaluating the current health status among a study population who presents with disease conditions.
In Q2, the item is very specific because it is referring to the fact that the respondent is feeling tired even at rest, which is not due to engaging in any form of strenuous physical exercise. Hence, both Q2 and Q3 shall elicit a response that has the same depiction of “being unhealthy”. In addition, based on content and statistical evidence (i.e Cronbach’s alpha and EFA), the two items are found to be fit to correlate with each other very well and should therefore be included in the same domain. Thank you.
Comment 2
Another instance is Q7 & Q8. Some people are lazy by nature. Movements of such a person may be slower than the same-aged people. However, this lazy person does not have any difficulty to move from one place to another.
Reply
To assess mobility, HRQ-6D has been designed to elicit the necessary responses from two items. The first item is related to the level of difficulty in making such a movement (Q7) and the second item is related to its intensity (Q8). Both items are considered essential for the scale of this HRQ-6D because there are instances in which some people do not experience any difficulty in making the movement but the intensity of such a movement can often be affected by other extrinsic factors such as aging or can be due to a particular neurological condition. On the other hand, the term “lazy” is quite general. For example, some people can be considered to be ‘lazy’ at work, but they are in fact very active in sports and other physical activities. Based on an assessment of their content validity and statistical evidence (i.e Cronbach’s alpha and EFA), the two items are found to be fit to correlate with each other very well and should therefore be included in the same domain. Thank you.
Comment 3
A complete statistical analysis was carried out in this study and the results are well interpreted.
Nonetheless, there is one unclear sentence. In the 2nd paragraph of the Sub-section 2.10, it is mentioned that ‘an Independent sample t-test, One-way Analysis of Variance test’ was applied. This statement needs further explanation and connects to the results.
Reply
Thank you for the compliment. The original statement is written as “Meanwhile, an Independent sample t-test, One-way Analysis of Variance test, and Pearson’s correlation test were also applied to determine the association or correlation between HRQ-6D and a differing set of conditions for an individual’s health status.”
The purpose of including the abovementioned statement in the main text of this paper is to ensure that the HRQ-6D scale does demonstrate an adequate level of association with an individual’s current health status. In order to determine such an adequate level of association between the two, the authors have actually applied the Independent sample t-test, One-way Analysis of Variance test, and Pearson’s correlation test to compare the HRQ-6D scores between a differing set of conditions of an individual’s health status. The results are presented in Table 5 (Independent sample t-test and One-way Analysis of Variance test) and Table 6 (Pearson’s correlation test). Thank you.
Round 2
Reviewer 2 Report
Use appropriate title(s) for Table 5.
Author Response
Dear Editor and Reviewer,
Noted. We revised the title as follows;
"Table 5: Association of HRQ-6D with the status of comorbidity and health conditions"
Thank you.
